# Valorization of Tomato Surplus and Waste Fractions: A Case Study Using Norway, Belgium, Poland, and Turkey as Examples

**DOI:** 10.3390/foods8070229

**Published:** 2019-06-27

**Authors:** Trond Løvdal, Bart Van Droogenbroeck, Evren Caglar Eroglu, Stanislaw Kaniszewski, Giovanni Agati, Michel Verheul, Dagbjørn Skipnes

**Affiliations:** 1Department of Process Technology, Nofima—Norwegian Institute of Food, Fisheries and Aquaculture Research, N-4068 Stavanger, Norway; 2ILVO—Institute for Agricultural and Fisheries Research, Technology and Food Science Unit, 9090 Melle, Belgium; 3Department of Food Technology, Alata Horticultural Research Institute, 33740 Mersin, Turkey; 4Department of Soil Science and Vegetable Cultivation, InHort—Research Institute of Horticulture, 96-100 Skierniewice, Poland; 5Consiglio Nazionale delle Ricerche, Istituto di Fisica Applicata ‘Nello Carrara’, 50019 Sesto Fiorentino, Italy; 6NIBIO—Norwegian Institute of Bioeconomy Research, N-4353 Klepp Stasjon, Norway

**Keywords:** tomato, valorization, sustainable production, processing, lycopene, waste reduction, vegetables, postharvest physiology, healthy food

## Abstract

There is a large potential in Europe for valorization in the vegetable food supply chain. For example, there is occasionally overproduction of tomatoes for fresh consumption, and a fraction of the production is unsuited for fresh consumption sale (unacceptable color, shape, maturity, lesions, etc.). In countries where the facilities and infrastructure for tomato processing is lacking, these tomatoes are normally destroyed, used as landfilling or animal feed, and represent an economic loss for producers and negative environmental impact. Likewise, there is also a potential in the tomato processing industry to valorize side streams and reduce waste. The present paper provides an overview of tomato production in Europe and the strategies employed for processing and valorization of tomato side streams and waste fractions. Special emphasis is put on the four tomato-producing countries Norway, Belgium, Poland, and Turkey. These countries are very different regards for example their climatic preconditions for tomato production and volumes produced, and represent the extremes among European tomato producing countries. Postharvest treatments and applications for optimized harvest time and improved storage for premium raw material quality are discussed, as well as novel, sustainable processing technologies for minimum waste and side stream valorization. Preservation and enrichment of lycopene, the primary health promoting agent and sales argument, is reviewed in detail. The European volume of tomato postharvest wastage is estimated at >3 million metric tons per year. Together, the optimization of harvesting time and preprocessing storage conditions and sustainable food processing technologies, coupled with stabilization and valorization of processing by-products and side streams, can significantly contribute to the valorization of this underutilized biomass.

## 1. Introduction

The tomato (*Solanum lycopersicum* (L.)), which is neither a vegetable nor a fruit but botanically speaking a berry, is currently spread across the world and is a key element in most cultures cuisines. The tomato originated in South America, from where it was imported to Mexico. Tomato came to Europe from the Spanish colonies in the 1500s along with several other “new” plants such as maize, potato, and tobacco. The tomato plant was immediately cultivated in the Mediterranean countries, but was initially poorly received further north in Europe. The skepticism of the tomato was due to that it was long suspected to be poisonous. As a curiosity, the tomato was not found in Norwegian grocery shelves until well into the 1950s, and was thus more exotic than oranges and bananas. Nowadays, however, tomatoes have definitely become an essential ingredient also in the North European and the Nordic cuisine. For example, in Norway, tomatoes have in recent years been the most sold product in the fresh vegetable segment, with a total turnover of approximately 15 million € and an annual consumption of 7.3 kg per capita, of which 1/3 is produced in Norway [1]. Including also processed tomato products, annual per capita consumption in Norway increases to 16.3 kg, whereas it is 23.5 and 27.5 kg in Poland and Belgium, respectively. These are however still low values compared to the Mediterranean diet; annual per capita consumption in Turkey, Armenia, and Greece is 94, 85, and 77 kg, respectively, whereas in Italy, Spain, Portugal, and Ukraine it is approximately 40 kg [2]. For a detailed list of European tomato production and consumption, see Appendix A. 

A joint FAO/WHO Expert Consultation report on diet, nutrition and the prevention of chronic diseases, recommended several years ago the intake of a minimum of 400 g of fruit and vegetables per day (excluding potatoes and other starchy tubers), for the prevention of chronic diseases such as cardiovascular diseases (CVD), diabetes, and obesity, as well as for the prevention and alleviation of several micronutrient deficiencies [3]. However, in the western world, the consumption of vegetables is still far less than recommended. There is thus a socioeconomic gain if one succeeds in stimulating increased consumption of tomato-based products high in lycopene and β-carotene that may lead to reduced incidents of cancers and CVD. To increase the consumption of vegetables, it is important to provide raw materials with high quality, diversity, and availability. Fruit and vegetables are important components of a healthy diet, and their sufficient daily consumption could help prevent major diseases such as CVD and certain cancers. According to the World Health Report 2002, low fruit and vegetable intake is estimated to cause ~31% of ischemic heart disease and 11% of stroke worldwide [4]. Overall, it is estimated that up to 2.7 million lives could potentially be saved each year if fruit and vegetable consumption were sufficiently increased. 

There is a large potential in Europe for optimization in valorization of crop biomass in the vegetable food supply chain. For example, there is occasionally overproduction of tomatoes for fresh consumption, and a fraction of the production is unsuited for fresh consumption sale (unacceptable color, shape, maturity, lesions, etc.). These tomatoes are normally destroyed and used as landfilling or animal feed, which represents an economic loss for producers and negative environmental impact. In Norway and Belgium, this surplus/waste fraction amounts to about 200 (Unpublished data from Rennesøy Tomat & Fruktpakkeri AS (2012), the biggest tomato packaging station in Norway) and 500 tons per year [5], respectively. A conservative estimate of € 4 per kg price increase for this raw material will thus yield a potential of 0.8 and 2.0 million € per year, respectively, for Norway and Belgium alone. Besides overproduction, part of the tomatoes produced in the greenhouse might not reach the market because they do not reach the local market standards. This can be due to cosmetic defects such as color, shape, size, etc. 

Additionally, there is loss of fresh tomato at the retailer’s level. In Norway, the general retailer loss is 10% for cluster tomatoes, 3–6% for single retail (“ordinary round”) tomatoes, and approximately 1% for cherry tomatoes, but it can be substantially higher in the peak of the growing season [6]. In the case of Norway, assuming a mean loss at retailers level of 5%, and that this loss can be halved by improved market regulation, there is a value increasing potential of a further 0.8 million € per year, again with a conservative price estimate of € 4 per kg. It is assumed that the total loss fraction is approximately the same in the other European countries, implying a much higher value potential in countries producing more tomatoes than Norway, considering that Norway is indeed a small tomato producer by European standards (Appendix A). An estimate for the total European market, based on a total waste fraction assuming a 15% waste in the tomato processing industry (including what is left in the field), and 5% waste in fresh market tomato, the waste fraction amounts to 3 million metric tons per year (Appendix A). The tomato processing waste quantities worldwide in 2010 was estimated to be between 4.3 and 10.2 million metric tons [7]. Based on this, it is beneficial to develop processing technology for the best possible utilization of this resource to improve economic sustainability of tomato production.

## 2. Tomato Production in Norway, Belgium, Poland and Turkey

### 2.1. Norway

Tomato production in Norway has been stable for the last 15 years with volumes between 9000 and 12,000 tons per year. Ninety percent of the production takes place in the county of Rogaland, on the southwest coast. Practically all tomatoes produced in Norway are destined for the domestic fresh market. The production is costly because the Norwegian climate necessitates the use of heated glass houses and artificial light for year-round production. Because of the high production costs, Norwegian tomatoes have traditionally not been subject to processing. Recently, some tomato farmers have found ways to alleviate the energy expenses by innovative solutions. One example of this is the ‘Miljøgartneriet’ which can be translated as ‘the Environmental plant nursery’ [8]. This glass house was built in 2010, covers 77,000 m^2^, and employs 70–85 workers in the high season. The innovations consist of amongst others the use of surplus CO_2_ and warm wastewater from a nearby dairy plant for plant feed and heating, respectively. Combined with other energy-efficient solutions in the construction of the glass house and recirculation of water, the production becomes more sustainable and with a low carbon footprint. An optimized year-round cultivation system achieved a yield of over 100 kg m^−2^ in commercial production, with an estimated maximum potential of 125–140 kg m^−2^ [9]. In Norway, the surplus fraction resulting from high-season overproduction amounts to 200 tons per year, corresponding to approximately 2% of total production. Total waste, i.e., combined with the waste at the retailer level and in the greenhouses, is estimated at ~6%. One of the main reasons for waste at the retailer level was found to be due to packaging. Comparing packaged cluster tomatoes to loose, unpacked single tomatoes revealed, contrary to expectation, that the former had significantly more wastage [6]. It was speculated that this was because packaging may lead to condensation and subsequent growth of molds. Therefore, packaging of warm tomatoes should be avoided [6]. Temperature abuse during transport and in the stores was also considered as main factors leading to waste [6].

Efforts have been made to produce tomato sauce of the surplus tomatoes. However, due to small volumes and high production costs, this turned out not to be economically viable and production stopped. At present, the fraction is primarily used as cattle feed, so that costs for disposal can be minimized. In order to overcome the seasonality problem, tomato surplus and waste fractions may be sorted and stored frozen in order to collect volumes for subsequent processing and perhaps to add this to batches of imported tomatoes for processing. A project is now starting up in Norway to look into this possibility for valorization and to identify and overcome the challenges related to this strategy.

### 2.2. Belgium

Belgian tomato production takes predominantly place in Flanders, where some 250 growers produced 220 to 260,000 tons per year on about 500 hectares between 2006 and 2016. Tomato is the second biggest crop under glass after lettuce, but generates the biggest economic return, ~180 million € per year, leaving the second and third place to strawberry and bell pepper, respectively. Belgian tomato growers deliver tomatoes for the internal fresh market during a period of about nine months each year. There are three areas in Flanders where the tomato growers cluster together, that is around Mechelen, Hoogstraten, and Roeselare. Roughly, half of the production is on the vine and the other half are loose tomatoes [10]. Also in Belgium, practically all tomatoes are produced in greenhouses and for fresh consumption. The average price the growers get for their tomatoes (loose and on the vine) is about € 0.75 per kg. As opposed to Norway, Belgian tomato export volumes are considerable. Today, approximately 70% of all tomatoes produced in Belgium are exported [10]. According to Lava, the cooperative of all Belgian fruit and vegetable auctions, the main exporting partners are France, Germany, the Netherlands, the UK, and the Czech Republic [11]. The tomato surplus fraction in Belgium makes up about 500 tons per year, corresponding to approximately 2% [5], as in Norway, and total waste (retailers, etc.) is estimated at approximately 5%. A recent study in Belgium estimated the losses of tomato that cannot be marketed due to cosmetic reasons to be ~1–2% only. Similar numbers were recorded for bell pepper and cucumber. This is very low compared to the percentages of other crops (e.g., zucchini 11.5% and lettuce 9.1%) [12]. 

### 2.3. Poland

Polish tomato production is different from Norwegian and Belgian production in several ways. First, the polish production volume is 3 times the Belgian and >60 times the Norwegian with a yearly production amounting to ~920,000 tons (~250,000 tons in the field and ~670,000 tons in greenhouses) [13], placing them among the top eight in Europe (Appendix A). Moreover, the production is carried out both in open field and under cover. The cultivation area under cover is approximately 27% of the total, but it is increasing [14]. Approximately 70% of the production takes place in Greater Poland, Kuyaivan-Pomeranian, Mazovian, and Switokrzyskie Provinces. Since Poland’s entry into the EU in 2004, fresh tomato exports have doubled, and accounts now for about 11% of production [14]. Approximately 1/3 of the total production is processed domestically, mainly into tomato paste and canned tomatoes (approximately 40,000 tons per year), and ketchup and tomato sauce (approximately 135,000 tons per year), whereof ~50% are exported [13]. The production of greenhouse tomatoes is intended for the fresh market and nearly 80% of production is sold on the internal market. The remaining 20% is export, and the main recipients are Ukraine, Belarus, the Czech Republic, Germany and the United Kingdom. The processing waste value can be assumed to be up to approximately 8.5%. This value consists of 1–3% seed waste, 2.8–3.5% skin, and up to 2% as whole fruit waste. Measures to reduce losses like choosing correct harvest time, avoiding damage during harvest, storage of crops protected from sunlight and immediate cool storage, the removal of damaged fruit, and using clean packaging material and proper transport are also important in Poland. The results obtained in open field tomato production in Poland depend very much on weather conditions. In some years, maturation is delayed and the quality of the fruits is poor, and it is very important to protect plants from diseases. These detrimental effects can be increased through improper nitrogen fertilization, which can delay the maturity of the fruits. In addition, the growing season in some years may be shorter due to the occurrence of early autumn frosts. In these conditions, unripen tomato fruits remain in the field and is lost. To reduce losses, proper nitrogen fertilization, early varieties with concentrated fruiting, and the use of ethylene to accelerate ripening are proposed.

### 2.4. Turkey

Turkey is the fourth largest tomato producer after China, India and the United States, yielding more than 7.2% of the world tomato production. The production amount was ~11.8 million tons in 2014 and 12.7 million tons in 2017. Sixty-seven percent of total production was evaluated as table tomato and 33% were industrially processed. More than 25% of the total production and 40% of table tomato is cultivated in greenhouses. Three-and-a-half-million tons (~28%) of the tomato production is being processed into paste, while 500,000 tons (4%) is used as sun-dried and canned (whole peeled, cubic chopped, puree, etc.). Due to the climate advantage, sun-dried tomatoes have great potential and almost all (97%) of them are exported. Tomato is the undisputed and clear leader product of the vegetable industry in Turkey. Tomato export is almost 40% of total fresh vegetable exportation of Turkey. From 10 to 18% of the total processing raw material is gone to waste [15]. Skin, seeds, fiber, etc. make up ~7% of this fraction, and the rest is mainly due to bad transportation in the tomato paste industry. Between 2010 and 2017, the average losses during harvesting were 3.5% and loss after harvesting was 10 to 15%. In 2017, pre- and postharvest losses were more than 2.1 million metric tons, corresponding to 16.5% of the total production [16]. Occurrences of exceptional high tomato wastage of up to 28% in specific regions have been reported [17]. Measures to reduce losses are summarized as choosing correct harvest time, avoiding damage during harvest, storage of crops protected from sunlight and immediate cool storage, the removal of damaged fruit, and using clean packaging material and proper transport [18]. The following precautions were proposed in order to reduce loss between harvesting and processing or wholesale: Choosing an earlier harvest time, using better packaging material at the farm stage instead of only traditional wooden or plastic cases, and refrigerated transport to the packaging or processing facilities [19]. 

## 3. The Significance of Lycopene in Tomato

Consumers are increasingly demanding naturally nutritious and healthy products that are produced without the use of genetic modification or additives and pesticide residues. It is therefore a large potential for developing processed products based on the part of the tomato production that does not go to fresh consumption. It turns out that the willingness to pay among the modern consumer increases when positive health effects attributed to the products can be documented. Most of the adult consumers are aware of the health benefits attributed to lycopene and other phytonutrients found in tomato, and thus lycopene is the second most important driver for consumer preferences, after price [20].

Lycopene is a member of the carotenoid family of compounds and is a key intermediate in the biosynthesis of many carotenoids. Lycopene is a pigment found in small amounts in many fruits and vegetables, and which, like carotene, gives rise to red color. Tomatoes are the main source of lycopene, while chili peppers may contain comparable amounts, and watermelon, red bell pepper, carrot, spinach, guava, papaya, and grapefruit contain relatively moderate amounts [21,22]. Lycopene occurs in several forms (isomers), some of which are taken up more easily by the human body than others [22,23,24]. The all-*trans* form is predominating in fresh tomato (~90%) [25], whereas it is the *cis* isomer that is most easily bioavailable to the human body [26,27,28]. Besides being an important nutrient, lycopene is also a very potent and sought after natural colorant with many applications in industrial food processing [29]. 

Research has shown that by means of processing it is possible to increase the proportion of the most bioavailable forms and stabilize these to thereby provide an increased health benefit. There is evidence that heat treatment and the addition of vegetable oils in tomato products increases the body’s absorption of lycopene compared with corresponding consumption of fresh tomato [25,27,30]. For lycopene to be absorbed in the duodenum, it must be dissolved in fat. The fat should not contain components which compete with lycopene for absorbing, such as vitamin E and K [25,31]. Although the biochemical mechanisms that make lycopene so beneficial to health are largely unknown, there is much to suggest that antioxidant and provitamin A properties can be crucial. 

### 3.1. Lycopene Content in Tomato

The lycopene range (0.03–20.2 mg/100 g) as reviewed in Table 1 is comparable to original results presented by Adalid et al. [32] where 49 diverse accessions of tomato from 24 countries on four continents displayed a span from 0.04 to 27.0 mg lycopene/100 g. In some wild species of tomato (*S. pimpinellifolium*), the lycopene concentration can be as high as 40 mg/100 g [33]. As shown in Table 1, the type and variety of tomato is also crucial for lycopene. Even the origin and the geographic location of their cultivation appears to play a major role [34,35]. This is probably due to different growing conditions and the degree of maturity [36,37], storage and transport conditions, etc. 

It is well known that tomato lycopene is concentrated in the skin and the water-insoluble fraction directly beneath the skin [53]. Table 2 demonstrates this partitioning and underpins that the tomato skin waste fractions is a good source of lycopene. Since lycopene and other carotenoids are most concentrated in and just inside the skin, lycopene is often higher per volume in small tomatoes of cherry type, because they have a relatively high peel to volume ratio. 

Since lycopene is the pigment responsible of the red hue of tomatoes, it can be derived that unripe tomatoes and light color tangerine varieties and green, orange and yellow varieties are lower in lycopene than mature red tomatoes [47,62]. Tomatoes with lower lycopene can be stored under special light and temperature so that they may accumulate lycopene before processing. Figure 1 illustrates the correlation between maturity stage, color, and lycopene content. 

Sikorska-Zimny et al., 2019 [66] proposed that, although tomatoes harvested at the full-ripe stage maintained 90% of their lycopene content for three weeks of storage, a compromise between firmness and storability may be found by harvesting at an earlier stage in order to balance the organoleptic and nutraceutical quality of the fruit. This means, at least for fresh-market tomatoes, that they can be harvested un-ripe in order to obtain storability without compromising neither on sensory or nutraceutical qualities, as long as proper storage conditions are obtained. 

### 3.2. Effect of Processing 

Processing strategies for tomatoes range from the very simple, as for fresh-market tomatoes, to complicated, as for the production of, e.g., tomato paste which includes multiple steps and several heat treatments such as drying, hot-break, and pasteurization [67]. Conclusions in relation to lycopene are that it is only slowly broken down by boiling (100 °C), and therefore constitutes no restriction for the heat treatment (Table 3) [67,68,69,70,71]. On the contrary, boiling for around two hours results in breakdown of carotenoid-associated protein structures so that lycopene is released, isomerization occurs and bioavailability increases [69,72]. Interestingly, Seybold et al., 2004 [70] found that lycopene isomerization occurred readily as an effect of thermal treatment in a standard lycopene solution, but this was not the case in tomatoes treated at the similar time/temperature conditions. Nevertheless, in freeze-dried lycopene powder, it was found that high temperatures (120 °C) and relatively short exposure time resulted in profound isomerization in both water and oil medium, but that loss of lycopene was significantly less in oil medium, presumably because oxidation was avoided [73,74]. Effects of thermal treatment on a range of health-beneficial antioxidants in tomato are reviewed in Capanoglu et al., 2010 [67], and it may seem that most of the other antioxidants (e.g., vitamin C and tocopherols, phenolics and flavonoids) are less heat-stable than lycopene. Mechanical and thermal treatment have significant effects on the consistency of tomatoes, the former mainly due to the release of pectin [75]. Mechanical treatment does not seem to affect the content of lycopene to any significant degree, but it may enhance bioavailability, especially when combined with thermal treatment [75]. Factors such as light, pH, and temperature is very critical to the stability of lycopene and carotenes [76]. Wrong processing or storage (i.e., exposure to light and oxygen) may, therefore, affect the ratio between isomers or totally degrade the beneficial compounds. However, when optimal storage criteria are met, lycopene is a very stable molecule [77]. Traditional processing methods have only little effect on the level of lycopene or isomerization [25]. In fact, thermal processing may generally increase the bioavailability of lycopene despite decrease of the total concentration of lycopene [27,57]. Studies that have followed the evolution of lycopene through the different processing steps of commercial tomato paste production are inconclusive, either reporting a small increase [41] or a significant decrease [78] as the tomatoes are processed into paste. Comparing rapid industrial scale continuous flow microwave pasteurization to conventional thermal processing of tomato juice, revealed that this novel energy-efficient technology resulted in a product with a higher antioxidant capacity and similar organoleptic, physiochemical and microbiological qualities [79]. High pressure processing (HPP) may increase lycopene extractability compared to conventional processing and result in higher carotenoid content, including lycopene, in tomato purées [80,81]. HPP also results in less lycopene cis-isomers compared to thermal processing [82]. 

Regarding lycopene and processing, the challenge is to limit the breakdown and stimulate the desired isomerization. In order to optimize the contents of isomerized lycopene, the kinetics of both isomerization and breakdown have to be known for the specific process. Experiments including a high number of time/temperature combinations should be done for a number of situations, e.g., aerobe vs. anaerobe processing and at different pH. It is only highly concentrated (e.g., dried powder) and, to a certain extent, concentrated and sterilized (canned) products that generally exhibit enlarged lycopene concentration compared to fresh tomato (Table 4 and Table 5). However, a heavy heat treatment is very energy intensive, and often leads to undesirable sensory properties. 

## 4. Utilization of Tomato Side Streams and By-Products

The valorization strategies for tomato waste biomass may be different depending on whether the primary production is originally intended for the fresh market or for industrial processing. For the former case, the biomass may mainly consist of surplus tomato due to seasonal overproduction or fractions perceived as unmarketable for cosmetics reasons, and in the latter of side streams and byproducts from the processing. Thus, the remainder of this chapter is divided into ‘Fresh tomato’ and ‘Processing’. However, the strategies described are not understood to be necessarily fixed in these categories, and can be interchanged (i.e., postharvest ripening can also be applied for processing tomatoes). Nevertheless, chosen strategies will depend on the available technologies and the volumes of the available biomass, and type of by-product/side stream fraction, which varies considerably in the countries subject to this case study.

### 4.1. Fresh Tomato

In Norway and Belgium, as mentioned above, domestically grown tomatoes are at present predominantly meant for the fresh market. Hence, processing by-products and side streams is not a big issue. However, mainly due to seasonal overproduction and, to a lesser extent, that a fraction of the tomatoes is not suitable for fresh market sale (wrong color, maturity level, shape, and injuries), there have been attempts to develop processing technology for this fraction. The valorization of this biomass is currently mainly impeded by the high moisture content and corresponding fast decay. The small volumes, geographical dispersity, and the seasonality make it even more challenging to process by conventional processing technologies. Alternatively, flexible and mobile processing technologies may be looked upon to valorize the underutilized tomato biomass. An example is the proposed novel spiral-filter press technology to refine horticultural by-products including tomato [86]. This technology alleviates the need of stabilizing the biomass by using expensive drying technology, and besides, it is flexible and may be used to produce a range of volumes as well as handle a multitude of different textures [87]. This implies that it may be used for, e.g., apple, berries, and carrot processing after the high-season tomato processing is over. 

Regards the surplus tomato fraction that is predominantly made up of unripe or underpigmented tomatoes, research has shown that these tomatoes can be turned into marketable tomatoes very effectively by simple means. In the SusFood1 era-net project ‘SUNNIVA’ [88], a range of elicitor treatments were tested in postharvest trials to identify efficient elicitor treatments as tools to influence the content of health-beneficial phytochemicals (HBPC) in tomato raw material and waste fractions. Products both for industry use and fresh market use were targeted. Results showed that the waste fractions of tomato could be utilized as valuable sources of HBPC, and also provide better raw material utilization when subjected to efficient postharvest elicitor treatments. Among the most promising elicitor treatments for tomatoes were ethylene treatments for pink and waste fractions (Figure 2). An important point of attention to maximize health benefits of industrial tomato products as well as tomatoes for fresh consumption is that different types or cultivars of tomatoes reach their maximum level of the HBPC at different maturity stages. 

Studies have shown that hormic dosage of ultraviolet radiation (UV-C) can be applied to delay the senescence of fruit and vegetables, suggesting that photochemical treatment may have the potential for postharvest preservation of tomato [90]. The effects of UV-C and temperature on postharvest preservation of tomato are summarized in Tjøstheim, 2011 [91], and long-term controlled atmosphere and temperature storage in Batu, 2003 [92] and Dominguez et al., 2016 [93]. In short, temperatures from 12.8 to 15 °C appear to be optimal, but there are large variations between different cultivars. An example of postharvest lycopene evolution in pink tomatoes at different storage temperatures is shown in Figure 3.

A completely different way of valorizing the fraction of tomatoes in the sub-optimal food (SOF) category, i.e., tomatoes with a color or shape that may be considered undesirable, is to target the consumers and try to get them more aware of the consequences of food waste. Consumers appear receptive to discounts on vegetables with imperfections [94]. Since October 2013, under its own brand “Wunderlinge” (translated as ‘odds’) such fruit and vegetables have been offered in Austria, and similar actions rapidly spread to neighboring countries [95]. Depending on season, and what is available, these fruits and vegetables, which, despite their idiosyncratic appearance is flawless in taste, are offered at a cheaper price. Similar campaigns and the establishments of ‘food-banks’ is becoming more common throughout Europe, but many actions are still at an experimental stage.

Then there will still be left fractions that are not suitable for recycling into the food chain. Upon extraction, both tomato fruit waste and vegetative by-products may be utilized as sources of compounds with pharmaceutical and therapeutic benefits (e.g., phenolic compounds like quercetins, kaempferol, and apigenin) or cosmetics ingredients (e.g., lactic acids) [96]. Side-flows and waste from vegetable processing can also be recirculated back to the field in the form of compost and used as growth substrates. Tomato waste compost may be used to replace partially peat-based substrate used for vegetable transplants production in nurseries [97]. Tomato side streams may also be used as raw material for the production of organic fertilizer or soil amendment. However, more research is needed to document the bio-stimulating effect of tomato waste streams for its potential use as an input source for such products [88].

### 4.2. Processing

During tomato processing, three to seven percent of the raw material is lost as waste [7,98]. The press cake resulting from tomato juice and sauce production consists of skin and seeds [99]. The seeds constitute approximately 10% of the fruit and 60% of the total waste, and is a source of protein and fat [100]. 

The chemical composition of tomato processing waste fractions was characterized by Al-Wandawi et al., 1985 [101]. The seed fraction was rich in oleic and palmitic acids, a high protein content with threonine and lysine as the dominating amino acids, and K, Mg, Na, and Ca as the dominating elements. Whereas the skin fraction was also rich in proteins with lysine, valine, and leucine as the predominating essential amino acids, and Ca, K, Na, and Mg as the major elements [101].

Pure lycopene has traditionally been extracted from tomatoes through processes using chemical solvents. Innovative supercritical fluid extraction (SFE) methods do not leave behind the chemical residues associated with other forms of lycopene extraction and were demonstrated by researchers at the University of Florida to be very efficient and with a greater yield than conventional methods [102]. Supercritical CO_2_ extraction using ethanol as a solvent is an efficient method to recover lycopene and β-carotene from tomato skin by-products [103]. Lenucci et al., 2015 [104] performed studies on enzymatic treatment of tomato biomass prior to supercritical CO_2_ extraction of lycopene, and the results showed that the enzymatic pretreatment could increase the yield of lycopene extraction by 153% as compared to solvent extraction. Besides enzymatic pretreatment, ultrasound and microwave-assisted extraction methods, on their own or combined, have been developed for the extraction of lycopene, resulting in higher extraction yield. Lianfu & Zelong, 2008 [105] compared combined ultrasound/microwave-assisted extraction (UMEAE) and ultrasonic assisted extraction of lycopene from tomato paste and achieved a yield of 97.4% and 89.4% for UMAE and UAE, respectively. UMEAE has thus shown to be highly effective and may also provide rapid extraction (367 s in the mentioned study [105]). The use of UAE was reviewed by Chemat et al., 2017 [106], who concluded that the process can produce extracts in concentrate form, free from any residual solvents, contaminants, or artifacts, and one of the most promising hybrid techniques is UMAE. Supercritical CO_2_ extraction has recently been optimized by modelling and resulted in a lycopene yield of 1.32 mg of extract per kg of raw material obtained by a peel/seed ratio of 70/30 [107], opening for a very promising future. Similarly, the use of pulsed electric fields (PEF) to improve carotenoid extraction from tomato was demonstrated [108]. The recent developments in carotenoid extraction methods was recently reviewed by Saini & Keum, 2018 [109], comparing enzyme-assisted extraction to the methods mentioned above and Soxhlet extraction. 

Lycopene is a high-value compound, costing on the order of 2000 €/kg in its pure form. Nevertheless, at a 10 mg lycopene per 100 g FW basis, it takes 10,000 kg of tomato fruit to produce 1 kg of pure lycopene even at 100% yield. Hence, in order to be economically sustainable, large volumes of the tomato raw material are needed. Consequently, this would hardly be feasible in Norway and Belgium, but may be proposed as a viable valorization strategy in countries like Poland and Turkey. Some strategies for valorization of tomato side streams and by-products are exemplified in Table 6.

## 5. Conclusions

Tomato side streams, by-products and surplus fractions are underutilized resources estimated to amount to in excess of 3 million metric tons per year in Europe. The ratios of this biomass that is consisting of whole fruit versus the processing side-streams are largely unknown. However, in regions where production is largely dependent on greenhouse production for fresh market sale due to climatic preconditions (e.g., Norway and Belgium), the fraction is predominantly whole fruit, and the opposite is the case where tomato processing constitutes a larger industry (e.g., Poland and Turkey). For the former case, strategies to prolong the postharvest storability of the fruit by, e.g., controlled atmosphere, elicitor, light, and temperature to overcome surplus tomatoes due to seasonal over production for the fresh market may be proposed, combined with novel, sustainable, low-energy, flexible processing technologies. For the latter case, where volumes of the side fractions make more sophisticated and targeted technologies economically and environmentally sustainable, several options for bioeconomical valorization exists, including utilization of by-products in comminuted hybrid and vegetarian food items, and the extraction of valuable health-beneficial compounds for the production of functional ingredients, protein-dense meals, and nutrient supplements. Strategies for utilization of inedible fractions, including the vegetative parts of the tomato plants may be found in the production of organic fertilizers, biobased materials such as paper, fiberboard, or extracts used in the cosmetics industries.

The notion that modern consumers are becoming more aware of the health beneficial properties of tomato and tomato products, and lycopene in particular, should not go unnoticed. Cultivation and processing practices may be further designed to meet consumer demands and preferences related to health and nutritional issues, and consequently add value to the tomato supply chain, also through the fabrication of functional and nutraceutical ingredients from biomass traditionally considered as waste. 

## Figures and Tables

**Figure 1 foods-08-00229-f001:**
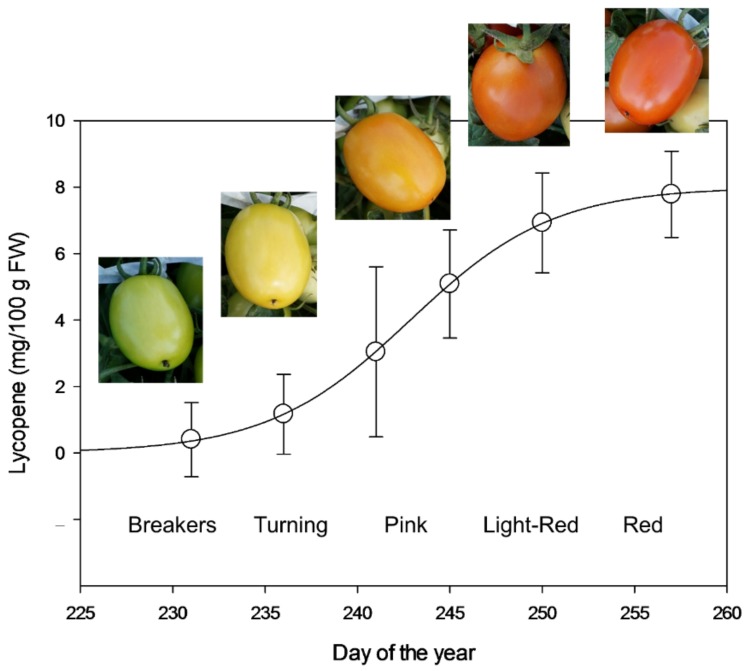
Lycopene evolution in processing tomatoes, cv. Calista, as measured during ripening in the field by a nondestructive optical method as previously described in Ciaccheri et al., 2019 [65]. FW: fresh weight.

**Figure 2 foods-08-00229-f002:**
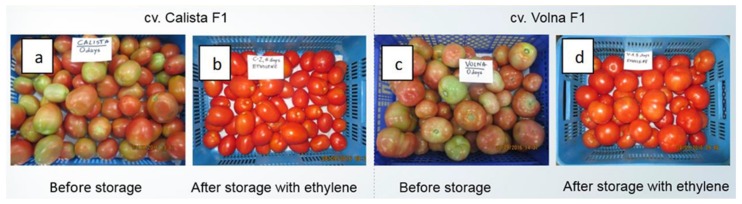
Example of ethylene treatment. Examples of Calista (**a**) and Volna (**c**) varieties that were not ripe at the time of harvesting and the respective varieties after six days of storage under ethylene atmosphere (**b**,**d**). Adapted from Grzegorzewska et al., 2017 [89], with permission.

**Figure 3 foods-08-00229-f003:**
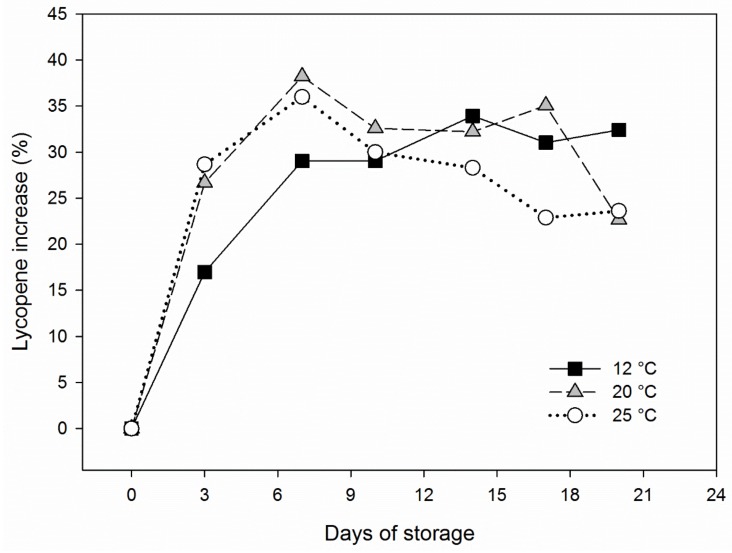
Lycopene increase (%) in pink harvested processing tomatoes, cv. Calista, during storage in the dark at 3 different temperatures (12, 20, and 25 °C) and 80% relative humidity. The initial level of lycopene was about 6.5 mg/100g fresh weight (FW). Rearranged from data previously published in Sikorska-Zimny et al., 2019 [66], with permission.

**Table 1 foods-08-00229-t001:** Lycopene content in tomato varieties (converted to mg/100 g fresh weight (FW)). Literature review. Values in italics are obtained by spectrophotometry, otherwise high-performance liquid chromatography (HPLC).

Variety	Total Lycopene	Type	Origin	Growth Conditions	Reference
Ministar	3.11	plum	SW Norway	Greenhouse, soil free	This study
Juanita	10.51	cherry			
Dometica	4.08	salad			
Volna	8.15	salad	Skierniewice, Poland	Field	
Calista	10.75	processing			
Pearson	*10.77*	N/A	California, USA	Field	[38]
DX-54	*~12*	N/A	Utah, USA	Field	[39]
Unknown	15.8	N/A	Florida, USA	Unknown	[40]
Amico	7.73	processing	Gödöllö, Hungary	Field	[41]
Casper	6.61				
Góbé	5.92				
Ispana	6.22				
Pollux	5.14				
Soprano	8.65				
Tenger	7.66				
Uno	7.09				
Zaphyre	6.95				
Draco	6.87				
Jovanna	11.61				
K-541	9.95				
Nivo	8.46				
Simeone	9.88				
Sixtina	10.51				
Monika	7.22	salad			
Delfine	6.51				
Marlyn	5.53				
Fanny	5.26				
Tiffany	6.23				
Alambra	5.40				
Regulus	6.59				
Petula	6.68				
Diamina	6.48				
Brillante	8.47				
Furone	5.18				
Linda	5.69				
Early Fire	*10.1–14.0*	processing		Field	[42]
Bonus	*8.5–12.7*				
Falcorosso	*8.0–11.1*				
Korall	*8.1–11.3*				
Nívó	*9.7–15.5*				
Strombolino	5.3–10.3	cherry, processing	Gödöllö, Hungary	Field	[43]
12 unnamed local varieties	*5.04–13.46*	N/A	SE Spain	Field	[44]
ACE 55 VF	*6.38*	Flattened globe			
Marglobe	*8.46*	Round			
Marmande	*7.01*	Flattened globe			
CIDA-62	6.23	cherry	Spain	Organic, field	[45]
CIDA-44A	2.95	round			
CIDA-59A	2.70				
BGV-004123	5.50				
BGV-001020	3.66	flattened and ribbed			
Baghera	4.64	round			
CXD277	15.33	processing	Spain	Field	[46]
H9661	12.21				
H9997	14.96				
H9036	11.36				
ISI-24424	17.01				
Kalvert	16.71				
Kalvert	20.2	processing	Lecce, Italy	Field	[47]
Hly18	19.5				
Donald	9.5				
Incas	9.3				
143	9.47	N/A	San Marzano, Italy	Field	[48]
Stevens	10.2				
Poly20	16.0				
Ontario	6.54				
Sel6	9.73				
Poly56	14.2				
1447	5.61				
977	10.0				
1513	9.21				
988	14.1				
Cayambe	13.5				
Heline	9.46				
1512	3.25				
1438	6.35				
Motelle	16.9				
Momor	13.3				
981	2.33				
Poly27	11.0				
Shasta	6.7–7.7	Early-season varieties	California, USA	Field	[49]
H9888	8.9–9.7				
Apt410	9.1–10.0				
CXD179	9.2–10.4	Mid-season varieties			
CXD254	10.5–12.0				
H8892	8.7–10.1				
CXD222	9.8–13.2	Late-season varieties			
H9665	8.7–12.2				
H9780	9.2–13.0				
Bos3155	14.92	Red varieties	California, USA	Field	[50]
CXD510	15.37				
CXD514	11.80				
CXD276	2.47	Light color Tangerine variety			
CX8400	0.08	Yellow variety			
CX8401	0.68	Orange variety			
CX8402	0.03	Green variety			
SEL-7	*3.23*	N/A	Haryana, India		[51]
ARTH-3	*4.03*				
Laura	12.20	N/A	New Jersey, USA	Greenhouse	[52]
Brigade	12.9	Processing	Salerno, Italy	N/A	[53]
PC 30956	18.7	High lycopene experimental hybrid,			
Cheers	*3.7*	N/A	Southern France	Greenhouse	[54]
Lemance	*3.7–6.9*	N/A	N/A	Greenhouse	[36]
Ohio-8245	9.93	Tomato pulp fraction	Ontario, Canada	N/A	[55]
92-7136	7.76				
92-7025	6.46				
H-9035	10.19				
CC-164	10.70				
Dasher	3.98	Plum	Italy	Greenhouse	[56]
Iride	4.45				
Navidad	4.89				
Sabor	5.22				
292	4.57				
738	4.77				
Cherubino	3.43	Cherry			
Crimson, green,	0.52	Salad	Ohio, USA	Purchased from local market	[57]
Crimson, breaker	3.84				
Crimson, red	5.09				
Unknown	10.14	Cherry	California, USA	Purchased from local supermarket	[58]
Unknown	5.98	On-the-vine			
Roma	8.98	Processing			
Jennita	1.60–5.54	Cherry	SW Norway	Greenhouse, soil free	[59] ^a^
Naomi	7.1–12.0	Cherry	Sicily, Italy	Cold greenhouse	[60] ^b^
Naomi	12.4–13.3	Cherry	Italy	Cold greenhouse	[34]
Ikram	8.5–8.9	Cluster			
Eroe	2.1–2.8	Salad			
Corbarino	6.8–14.6	Cherry	Battipaglia, Italy	Field grown	[61] ^c^

(a) Harvested twice monthly from May to October. (b) Harvested at six different times throughout the year. (c) As an effect of N and P fertilization load.

**Table 2 foods-08-00229-t002:** Lycopene content in peel versus pulp in some tomato varieties (converted to mg/100 g FW). Literature review. Values in italics are obtained by spectrophotometry, otherwise HPLC.

Cultivar	Total Lycopene (Converted to mg/100 g FW)		Comment	Reference
	Peel	Pulp		
**HLT-F61**	89.3	28.0	Field grown, Northern Tunisia	[62]
**HLT-F62**	50.8	16.7		
**Rio Grande**	42.4	10.1		
**8-2-1-2-5**	*14.3*	*6.7*	Harvested at mature green stage (Ludhiana, India) and stored at 20 °C until ripe	[37]
**Castle Rock**	*13.1*	*6.2*		
**IPA3**	*10.2*	*4.0*		
**Pb Chhuhra**	*8.6*	*4.6*		
**UC-828**	*6.5*	*3.7*		
**WIR 4285**	*6.5*	*3.1*		
**WIR-4329**	*8.1*	*4.3*		
**818 cherry**	*14.1*	*6.9*	Field grown, New Delhi, India	[63]
**DT-2**	*8.1*	*5.2*		
**BR-124 cherry**	*10.2*	*4.9*		
**5656**	*10.7*	*4.5*		
**7711**	*9.0*	*4.4*		
**Rasmi**	*10.8*	*4.3*		
**Pusa Gaurav**	*10.2*	*4.0*		
**T56 cherry**	*12.0*	*3.8*		
**DTH-7**	*4.8*	*2.7*		
**FA-180**	*7.6*	*2.5*		
**FA-574**	*6.1*	*2.2*		
**R-144**	*6.2*	*2.0*		
**Grapolo**	6.0	1.2	Purchased in supermarket or open-air market, Zagreb, Croatia,	[64]
**Italian cherry tomato**	7.2	2.0		
**Croatian cherry tomato**	5.3	1.6		
**Croatian large size tomato**	3.5	1.3		
**Turkish large size tomato**	3.3	1.2		

FW: fresh weight.

**Table 3 foods-08-00229-t003:** Effects of heat treatment on tomato products.

Processing	Heat Treatment	Effect	Texture	Taste	Lycopene	Color
Chopping raw	Mild < 80 °C	Enzymes are released, pectin degraded and hexanal/hexanol formed	Thick before heating, then soup	Vivid Green	Unchanged	Poor, controlled by pH
	Strong > 80 °C			Moderate green	Increased	Acceptable
Chopping raw, waiting for thickening before cooking, for example 2 h	Instantly to 100 °C, medium shortly, to thicken		Slightly thick and thickens with increased cooking time	Vivid Green	Somewhat increased	Acceptable
Chopping cooked	Mild < 80 °C	Enzymes inactivated only partially	Thin	Moderate green	Unchanged	Poor, controlled by pH
	Strong > 80 °C	Enzymes inactivated	Thick	Green aroma	Unknown	Acceptable
Puree, unpeeled	2 h 100 °C	Carotenoid content maximum after 2 h.	Unknown	A little green	Most after 2 h *	Most after 2 h
Puree, peeled		Carotenoid content low and stable unaffected by time			Less than unpeeled	Less than unpeeled

* Carotenoid associated protein structures are broken down so that lycopene is released and isomerization occurs so that the bioavailability increases.

**Table 4 foods-08-00229-t004:** Lycopene content in tomato products (converted to mg/100 g FW). Literature review. Values in italics are obtained by spectrophotometry, otherwise HPLC.

Product	Total Lycopene	Comment	Reference
Pulp	10.6–18.7	Commercial products, Salerno, Italy	[53]
Purée	12.7–19.6		
Paste	57.87	Commercial products, California, USA	[58]
Purée	23.46		
Juice	10.33		
Ketchup	12.26–14.69		
Juice, heat concentrated	2.34	Experimentally processed from Crimson-type tomatoes purchased from local markets, Ohio, USA	[57]
Paste, heat concentrated	9.93		
Soup, retorted	10.72		
Sauce, retorted	10.22		
Juice	7.83	Experimentally processed from tomatoes purchased from local markets and heat treated according to standardized industrial food processing requirements	[25]
Soup, condensed	7.99		
Canned whole tomato	11.21		
Canned pizza sauce	12.71		
Paste	30.07		
Powder, spray dried	126.49		
Powder, sun dried	112.63		
Sun dried in oil	46.50		
Ketchup	13.44		
Tangerine tomato sauce	4.86	Experimentally processed from tomatoes grown at the Ohio State University, USA	[30]
Tangerine tomato juice	2.19		
Red tomato juice	7.63		
Regular salad tomatoes, Gran Canaria, Spain	1.15	Fresh	[69]
	1.09	Boiled 10 min	
	0.99–1.18	LTLT, 60 °C 40 min	
	1.07–1.23	HTST, 90 °C 4 min	
Bella Donna on the vine, Netherlands	3.80	Fresh	
	3.06	Boiled 20 min	
	3.91–4.31	LTLT, 60 °C 40 min	
	3.43–4.15	HTST, 90 °C 10 min	
Daniella, Spain	2.37	Fresh purée	[81]
Daniella, Spain	0.99	Fresh	[80]
	1.48	HP (400 MPa, 25 °C, 15 min)	
	0.86	Pasteurization (70 °C, 30 s)	
	0.95	Pasteurization (90 °C, 60 s)	
Torrito, Spain	39.67	Fresh	This study
	26.39	HTST (90 °C, 15 min)	
	23.77	HP (400 MPa, 90 °C, 15 min)	
Torrito, the Netherlands	11.44	Fresh	
	7.57	HTST (90 °C, 15 min)	
	10.10	HP (400 MPa, 90 °C, 15 min)	
	10.00	HP (400 MPa, 20 °C, 15 min)	
	5.41	HP (600 MPa, 90 °C, 15 min)	
	4.08	HP(600 MPa, 20 °C, 15 min)	
Heinz purée, USA	6.62	Puré, fresh	[83]
	6.61	Boiled 5 min	
	6.57	Boiled 10 min	
	6.48	Boiled 30 min	
	6.39	Boiled 60 min	
Double concentrated commercial canned tomato purée, Netherlands	*39*	Unheated	[68]
	*31*	Autoclaved 100 °C, 20 min	
	*29*	Autoclaved 100 °C, 60 min	
	*29*	Autoclaved 100 °C, 120 min	
	*28*	Autoclaved 120 °C, 20 min	
	*30*	Autoclaved 120 °C, 60 min	
	*29*	Autoclaved 120 °C, 120 min	
	*31*	Autoclaved 135 °C, 20 min	
	*33*	Autoclaved 135 °C, 60 min	
	*32*	Autoclaved 135 °C, 120 min	
Experimental purée	3.79	Unheated	[84]
	5.93	Steam retorted, 90 °C, 110 min	
	5.20	Steam retorted, 100 °C, 11 min	
	4.74	Steam retorted, 110 °C, 1.1 min	
	3.37	Steam retorted, 120 °C, 0.11 min	
FG99-218, USA	16.04	Juice, fresh	[85]
	16.05	Juice, hot break	
	17.95	Juice, HP (700 Mpa/45 °C/10 min)	
	17.12	Juice, HP (600 Mpa/100 °C/10 min)	
	15.50	Juice, TP (100 °C/35 min)	
OX325, USA	9.84	Juice, fresh	
	10.22	Juice, hot break	
	10.88	Juice, HP (700 Mpa/45 °C/10 min)	
	10.29	Juice, HP (600 Mpa/100 °C/10 min)	
	8.49	Juice, TP (100 °C/35 min)	

LTLT: Low Temperature, Long Time; HTST: High Temperature, Short Time; HP: High Pressure processing.

**Table 5 foods-08-00229-t005:** Lycopene content (mg/100 g FW) in tomato and tomato products including the fractions of *trans*- and *cis*-isomers.

Produkt	Total Lycopene	All *Trans* Lycopene (% of Total)	*Cis* Lycopene (% of Total)	Reference
Conesa tomato paste, Spain, 0.16% fat Batch 1 (2014)	32.1	29.2 (91.0)	2.9 (9.0)	This study
Conesa tomato paste Spain, 0.16% fat Batch 2 (2015)	26.6	23.6 (88.7)	3.0 (11.3)	
Conesa tomato paste Spain, 0.16% fat Batch 2 (2015)—Autoclaved	22.8	19.9 (87.3)	2.9 (12.7)	
Conesa tomato paste Spain, 0.16% fat Batch 2 (2015)—Microwaved	22.9	20.1 (87.8)	2.8 (12.2)	
Conesa tomato fine chopped, Spain, 0.04% fat	6.5	5.9 (90.8)	0.6 (9.2)	
Heinz ketchup 0.1% fat	11.0	9.4 (85.5)	1.6 (14.5)	
Eldorado tomato puree, Italy, 1% fat	32.1	29.4 (91.6)	2.7 (8.4)	
Cherry tomatoes	10.14	8.91 (87.9)	1.23 (12.1)	[58]
On-the-vine tomatoes	5.98	5.00 (83.6)	0.98 (16.4)	
Roma tomatoes	8.98	7.88 (87.7)	1.10 (12.3)	
Tomato paste	57.87	45.94 (79.4)	11.93 (20.6)	
Tomato purée	23.46	17.85 (76.1)	5.61 (23.9)	
Tomato juice	10.33	8.47 (82.0)	1.86 (18.0)	
Tomato ketchup	12.26–14.69	9.40–9.47 (64.4–76.7)	2.86–5.22(23.3–35.6)	

**Table 6 foods-08-00229-t006:** Proposed utilization of tomato side streams and by-products from food processing.

Product	Active Ingredients	Fraction	Reference
Color pigments, Antioxidants	Lycopene	Skin, pomace, whole fruit	[72,102,110]
Tomato seed oil	Unsaturated fatty acids (linoleic acid)	Seeds	[111,112]
Thickening agent	Pectin	Dried Pomace	[113,114]
Comminuted and vegetarian sausages	Dried and bleached tomato pomace	Dried Pomace	[115]
Tomato seed meals	Protein, polyphenols, etc.	Seeds, pomace	[116]
Nutrient supplements	Vitamin B12	Pomace	[117]
Cosmetics	Phenolic compounds, antioxidants, lactic acid, etc.	Whole plant	[96]
Compost, growth substrates, fertilizer	Phytochemicals	Whole plant	[97]

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
