# Peer review of "Valorization of Tomato Surplus and Waste Fractions: A Case Study Using Norway, Belgium, Poland, and Turkey as Examples"

_foods, 2019, doi:10.3390/foods8070229_

Round 1
Reviewer 1 Report
The article author(s) may improve on the following:
Abstract: does not have conclusion and recommendation
Keywords: Not included
Include scientific name and authority at the first mention of the crop
Conclusion: The author(s) give clear conclusion and future recommendations based on the study. This does not come out boldly from the conclusion
Line 400 to 402: need to be addressed
Author Response
Answers to reviewer 1
Comment 1: Abstract: does not have conclusion and recommendation
Answer 1: A short paragraph highlighting the key conclusions has been added (highlighted in yellow).
Comment 2: Keywords: Not included
Answer 2: Key words are included in the revised version.
Comment 3: Include scientific name and authority at the first mention of the crop
Answer 3: This is included in the revised version.
Comment 4: Conclusion: The author(s) give clear conclusion and future recommendations based on the study. This does not come out boldly from the conclusion
Answer 4: It is not clear what the Reviewer means by this comment. We have tried to condense the most important findings in the conclusion and also in the revised Abstract.
Comment 5: Line 400 to 402: need to be addressed
Answer 5: This is addressed in the revised version
Reviewer 2 Report
Valorization of tomato surplus and waste represents a great interest in recent years. The manuscript is very well structured. The information provided is very useful from a practical point of view. It is significant in content. The article will be in high interest to the readers. The introduction provides sufficient background and describes clear the main goal of the present research. The references used are sufficient in number and cover the available scientific and practical information on the topic. The available data are analyzed, interpreted and critical reviewed in detail and correctly. Conclusion highlights the main aspects on the study.
1) Rows 34-35: The keywords are missing.
2) Row 244: The year of publication (Seybold et al.) should be cited.
3) It would be better to note in Section 4.2 other available effective lycopene extraction technologies such as enzyme and ultrasound assisted extraction, together with supercritical CO2 extraction.
Author Response
Answers to reviewer 2
Comment 1: Rows 34-35: The keywords are missing
Answer 1: Key words are included in the revised version.
Comment 2: Row 244: The year of publication (Seybold et al.) should be cited.
Answer 3: This is included in the revised version.
Comment 3: It would be better to note in Section 4.2 other available effective lycopene extraction technologies such as enzyme and ultrasound assisted extraction, together with supercritical CO2 extraction.
Answer 3: This section has been expanded as suggested by Reviewer 2 in the revised version. We also opt to refer a recent review paper on the topic (Saini & Keum, 2018).
Reviewer 3 Report
Overall comment:
The paper is interesting and presents a review of tomato from two perspectives, valorization of tomato surplus and preservation and enrichment of lycopene. However, having these two perspectives is also the downfall of this manuscript: The connection between these two perspectives is not clear and must be approved prior publication. The impression after reading the paper is that it consists of two sections that are not in the same level of detail. The lycopene chapter has very detailed information and chapter 4 covers multiple pages of the manuscript. However, in the Abstract and Conclusions, lycopene is hardly mentioned. The tomato waste/surplus chapters are adequate in the level of detail, however, more careful use of referencing is needed.
Introduction in general:
It is not well presented, what is the aim of this paper
Norway especially is highlighted in the Introduction. Some mentioning of Belgium. However, the title of the manuscript leads to believe that it will be balanced examination of these four countries. Please clarify.
p3, para3;
How is the amount and costs of surplus tomatoes defined in Norway and Belgium? The links provided do not give the data.
As Supplementary material was not provided for review, if I am not mistaken, it is not possible to evaluate the relevance of the amount of wasted tomatoes in European level.
Chapter 2:
Are there imported tomatoes used in these countries? I would assume so, at least Norway.
A map of Europe indicating the production areas (Rogaland, Flanders etc) and exported tomato steams to other countries would be helpful
No references are provided for tomato surplus amounts or total waste percentages in countries
The information provided is not consistent between the four countries. E.g. for Poland and Turkey the tomato products are described, but not for other countries
Why measures and precautions to reduce food losses are under Turkey? Are they not relevant for other countries as well?
Chapter 3:
It is not self-evident to the reader, what is the meaning of this chapter in this manuscript? If it is important, please think of more reflective title than “Lycopene” and introduce the importance and relevance more clearly in the start of the paragraph.
Please define HPLC
I am not an expert on lycopene. With that in mind, I feel the reader is left alone to analyze these huge tables in chapter 3.
Are there differences between the four countries in terms of lycopene?
Chapter 4:
Consider changing the titles “4.1. Fresh tomato” and “4.2. Processing” to be more descriptive
Author Response
Answers to reviewer 3
Comment 1: It is not well presented, what is the aim of this paper. Norway especially is highlighted in the Introduction. Some mentioning of Belgium. However, the title of the manuscript leads to believe that it will be balanced examination of these four countries. Please clarify.
Answer 1: In the introduction, the rising popularity of the tomato among customers is exemplified by Norway. The valorization potential of surplus fractions of tomatoes intended for the fresh market is exemplified by Norway and Belgium (for which the background data are more certain), with the implication of a high value increasing potential also in the other tomato producing countries. However, Section 2 is dedicated to an examination of the four countries, and this section is edited in the revised version to make it more balanced.
Comment 2: How is the amount and costs of surplus tomatoes defined in Norway and Belgium? The links provided do not give the data.
Answer 2: We regret that the provided links did not support our claims, and this is corrected in the revised manuscript. It is indeed not straightforward to find good written documentation for these estimates. However, regarding the surplus fraction in Norway: Data from Rennesøy Tomat & Fruktpakkeri AS (2012), the biggest tomato packaging station in Norway estimates this fraction to 200 tons per year, corresponding to approx. 2% of total production. The total waste fraction (including also waste at retailers level) is estimated to 6% based on the referred report by Vold et al., 2006 (in Norwegian) concluding that the waste at retailers level of Norwegian produced tomatoes (not including import) was stable from 1996 to 2006. For Belgium, we have referred to an ILVO report (Kips & Van Droogenbroeck, 2014, in Dutch), and Ghellynch et al. 2017 (also in Dutch). We regret that these references are not easily accessible and in English. However, this fact underlines the novelty and pertinence of our study.
The price of fresh tomatoes in Norway varies from 17 to 200 NOK per kilo, corresponding to 1.85 to 21.7 € per kilo with today’s currency exchange rate. Norwegian produced cherry tomatoes are in the upper price range with prices of typically 15-20 € per kilo, while imported ordinary single round type tomatoes are the cheapest. Production volumes in Norway can be divided into ‘special tomatoes’ which is on-the-vine, cherry, plum, etc. constituting ~60% of production. The mean retail price of this category is approx. €10 per kilo, compared to approx. €7 per kilo for the remaining 40%. On-the-vine tomatoes dominates the waste fraction (>50 %). Considering also that the price value of this fraction ‘as is’, is ≥ zero (as cattle feed), or even negative if costs for disposal applies, the net value increase of €4 per kg is indeed conservative. Admittedly, the price of tomatoes is lower in Belgium and in other parts of Europe than in Norway, so from the above reasoning the estimates concerning Belgium may seem over estimated. However, the costs for disposal of waste is considerably higher in less rural regions, so when such expenses are included (as it should be when calculating value increase), the estimate is reasonable also concerning Belgium.
Comment 3: As Supplementary material was not provided for review, if I am not mistaken, it is not possible to evaluate the relevance of the amount of wasted tomatoes in European level.
Answer 3: It is a pity that the Supplementary material was not provided for review. It was uploaded to the submission platform. In the Supplementary Table, estimated waste is provided for the five top producing countries (with Turkey as no. 1) and includes in addition Poland, Belgium and Norway.
Comment 4: Are there imported tomatoes used in these countries? I would assume so, at least Norway.
Answer 4: Net export for all tomato producing countries in Europe is provided in Supplementary material. The import to Norway is ~68 000 tons per year
Comment 5: A map of Europe indicating the production areas (Rogaland, Flanders etc) and exported tomato steams to other countries would be helpful
Answer 5: The authors find that a map of Europe indicating the production areas will not be efficient. Readers with special interest on this issue can readily find this information elsewhere. Export streams for the four case countries are indicated in the text.
Comment 6: No references are provided for tomato surplus amounts or total waste percentages in countries
Answer 6: Sources (when available) for total waste are provided in Supplementary material. This includes the top European tomato producing countries and the case countries studied. We have included also more references in the main text concerning the case countries. Again, we apologize that relevant references are not easily accessible available in the international literature. However, all authors are experts in the field and is in regular contact with the industry and primary producers in their respective countries, and have collected reliable and up-to-date data concerning the case countries. Regarding references for total production and consumption in all European countries, we want to point out that these are gathered from the recognized FAOSTAT (2012) database freely accessible through http://chartsbin.com/view/32687 (reference no. 2) and only supplemented with more recent data when available. When performing estimates on waste, these background data have, to varying extent, formed the base when empirical data was missing. This is clearly stated in Supplementary material.
Comment 7: The information provided is not consistent between the four countries. E.g. for Poland and Turkey the tomato products are described, but not for other countries
Answer 7: As stated clearly in the original manuscript, at present, practically all production in Norway and Belgium is intended for fresh market sale of whole tomatoes. This means that other (processed) products are very marginalized and only on an extreme niche scale or an experimental stadium, and are thus not produced in volumes relevant to mention beyond what is already done in the original manuscript.
Comment 8: Why measures and precautions to reduce food losses are under Turkey? Are they not relevant for other countries as well?
Answer 8: We agree that this is a flaw in our original manuscript and has corrected it accordingly.
Comment 9: Chapter 3: It is not self-evident to the reader, what is the meaning of this chapter in this manuscript? If it is important, please think of more reflective title than “Lycopene” and introduce the importance and relevance more clearly in the start of the paragraph.
Answer 9: We agree to this comment and have revised accordingly.
Comment 10: Please define HPLC
Answer 10: High-performance liquid chromatography (HPLC). This is spelled out at first encounter in the revised manuscript.
Comment 11: I am not an expert on lycopene. With that in mind, I feel the reader is left alone to analyze these huge tables in chapter 3.
Answer 11: Re. Table 1: Although we admit that it may not be straightforward to compare lycopene content obtained in different regions around the globe, different climatic zones, different varieties, under cover and in the field, different irrigation and fertilization etc., we feel that the inclusion of this table is justified. Whenever lycopene is regarded the number one sales argument for tomato and tomato products, Table 1 may serve as a first indicator for selection of which varieties to choose for cultivation if a high lycopene content is desired. It also serves as documentation for the assertions put forward in the text. Table 2 is a very effective demonstration that the peel fraction is high in lycopene compared to the pulp fraction, and that this can be considered a general trait. Generally, the tables are included to demonstrate the claims put forward in the text. Table 4 is a comprehensive review on the effects of different processing designs on lycopene content in tomato products, commercial as well as experimental. This table also contains some original data on HPP treatment, demonstrating the significance of pressure level and time, which is scarce in international literature. Table 5 shows the distribution between cis and trans lycopene in processed products. This is well known in fresh tomato, but not to the same extent in processed products. This table is also 50% original data. Both Table 4 and 5 serves as a follow up and an empirical testimony to the more general approach presented in Table 3.
Comment 12: Are there differences between the four countries in terms of lycopene?
Answer 12: We do not have a big enough collection of data to answer this question. However, as stated in the paper; origin and geographic location of cultivation appears to play a major role. It is generally accepted that there is variability in lycopene content between different varieties of tomato, but as demonstrated in the present paper, the same varieties also seem to differ in lycopene content depending on origin. Additionally, the selection of cultivars is different in the four countries, probably due to tradition, culture, and the fact that cultivars are adapted to the highly differing climatic conditions in the countries studied.
Comment 13: Chapter 4: Consider changing the titles “4.1. Fresh tomato” and “4.2. Processing” to be more descriptive
Answer 13: We acknowledge this suggestion, and opted for a brief introduction to explain the choice of titles in these sections.
Round 2
Reviewer 1 Report
no further comments
Reviewer 3 Report
Lycopene still is mentioned only briefly in the abstract, which I find as a shortcoming. However, the authors have provided sufficient response to my comments, so I have no further comments.